# Enzymatic Synthesis of a Novel Coumarin Aminophosphonates: Antibacterial Effects and Oxidative Stress Modulation on Selected *E. coli* Strains

**DOI:** 10.3390/ijms24087609

**Published:** 2023-04-20

**Authors:** Dominik Koszelewski, Paweł Kowalczyk, Anna Brodzka, Anastasiia Hrunyk, Karol Kramkowski, Ryszard Ostaszewski

**Affiliations:** 1Institute of Organic Chemistry, Polish Academy of Sciences, Kasprzaka 44/52, 01-224 Warsaw, Poland; anna.brodzka@icho.edu.pl (A.B.); anastasiia.hrunyk@icho.edu.pl (A.H.); ryszard.ostaszewski@icho.edu.pl (R.O.); 2Department of Animal Nutrition, The Kielanowski Institute of Animal Physiology and Nutrition, Polish Academy of Sciences, Instytucka 3, 05-110 Jabłonna, Poland; 3Department of Physical Chemistry, Medical University of Bialystok, Kilińskiego 1 Str., 15-089 Białystok, Poland; kkramk@wp.pl

**Keywords:** aminophosphonates, enzymatic catalytic promiscuity, multicomponent reaction, coumarins, Kabachnik-Fields reaction, lipases, antimicrobial activity, *E. coli* cells, MIC

## Abstract

The objective of the present study was to evaluate the synergistic effect of two important pharmacophores, coumarin and α-amino dimethyl phosphonate moieties, on antimicrobial activity toward selected LPS-varied *E. coli* strains. Studied antimicrobial agents were prepared *via* a Kabachnik–Fields reaction promoted by lipases. The products were provided with an excellent yield (up to 92%) under mild, solvent- and metal-free conditions. A preliminary exploration of coumarin α-amino dimethyl phosphonate analogs as novel antimicrobial agents was carried out to determine the basic features of the structure responsible for the observed biological activity. The structure–activity relationship revealed that an inhibitory activity of the synthesized compounds is strongly related to the type of the substituents located in the phenyl ring. The collected data demonstrated that coumarin-based α-aminophosphonates can be potential antimicrobial drug candidates, which is particularly crucial due to the constantly increasing resistance of bacteria to commonly used antibiotics.

## 1. Introduction

*Escherichia coli* is a gram-negative bacteria which assists as a diverse microorganism in the fields of biotechnology and microbiology, where it has served as the host organism for the majority of work with recombinant DNA. This normally harmless commensal may easily become a highly adaptable pathogen causing serious diseases, such as gastroenteritis and extraintestinal infections of the urinary tract, bloodstream, and central nervous system [1]. The worldwide consequences of these infections are extraordinary, as hundreds of millions of people suffer from them [2]. Multi-drug resistance of *E. coli* to antimicrobial medicines, especially in developing countries, is considered one of the main causes of an ineffectiveness in the treatment of infectious diseases [3]. According to the World Health Organization (WHO) reports, multi-drug resistant pathogens are among the biggest challenges in treating bacterial infections worldwide, causing 10 million deaths a year by 2050 [4]. As a result of these premises and due to the high degree of antimicrobial resistance, it is necessary to develop effective antimicrobial drugs to treat bacterial resistance [5]. Current strategies in the pharmacological research of new lead compounds mostly refer to a large collection of molecules proven to be broadly useful as therapeutic agents, such as coumarins. Therefore, compounds that contain the 2*H*-chromen-2-one structural motif are well known in a variety of natural products [6,7,8,9]. Numerous biological activities have been attributed to simple coumarins as well as pyranocoumarins, furocoumarins, and analogues, e.g., antimicrobial, antiviral, anticancer, enzyme inhibition, anti-inflammatory, antioxidant, and central nervous system activities [10,11,12,13,14,15,16,17]. Coumarin and its derivatives found an important class of heterocyclic compounds that hold a dominant place in medicinal chemistry [18] (Figure 1).

In the last decade, α-aminophosphonates were intensively studied as a significant group of compounds because of their high biological activity [19,20,21], which broadens their applicability in agricultural and medicinal chemistry. Organophosphorus compounds are known for their antimicrobial activity (alafosfalin); for example, dialkyl α-aminophosphonates exhibit activity against pathogenic *E. coli*, *S. aureus*, *Bacillus*, or *K. pneumonia* strains (Figure 1) [22,23,24,25]. Based on the few available literature reports, the synergetic effect of these two pharmacophores, coumarin and α-aminophosphonate, incorporated in the same structure of the compound, may provide the desired antimicrobial effect, providing highly effective compounds against drug-resistant *E. coli* pathogenic strains [26,27,28,29]. 

Our previous studies revealed that several aminocoumarin peptidomimetics as well as α-aminophosphonates were effective inhibitors of growth of various LPS *E. coli* strains [30,31,32]. Thus, the aim of the current work is to develop a metal-free protocol for coumarin α-amino phosphonate derivatives preparation aimed at identifying features of the structure that could be important for anti-*E. coli* activity (Figure 1).

## 2. Results and Discussion

### 2.1. Chemistry

The Kabachnik–Fields reaction is the most practical approach for α-aminophosphonate preparation, as widely reported in the literature [33,34]. To improve the reaction yield, numerous protocols have been elaborated, such as ultrasound irradiation [35], the usage of ionic liquids [27], synthesis on the solid phase by mechanochemistry [36], and microwave irradiation [37]. The reactions can be mediated by various catalysts, such as SnCl_2_ [38], TiO_2_ [39], FeCl_3_ [40], Yb(PFO)_3_ [41], SbCl_3_/Al_2_O_3_ [42], and CF_3_CO_2_H [43]. However, a majority of them are toxic, expensive, and have to be used in stoichiometric amounts. Moreover, reported methods suffer from some drawbacks, such as moderate yields, tedious product isolation, and the use of unrecyclable catalysts. Additionally, due to pharmaceutical requirements, limits of heavy metal must be maintained below 5 ppm, which makes the above mentioned catalysts hardly acceptable for the industries. The green approaches toward α-aminophosphonates are rather limited [44]. Enzymes, which are natural catalysts with high catalytic activity, appear to be the unsurpassed alternative for the elaboration of new synthetic protocols that fulfill the restrictions related to safety and environmental protection. What is more, enzymes enable the synthesis of compounds without metal contamination, which is especially esteemed by the pharmaceutical industries.

Regarding non-natural activity of lipases [32], the model multicomponent reaction of 7-amino-4-(trifluoromethyl)coumarin (1 mmol), benzaldehyde (1 mmol), and dimethyl phosphite (1 mmol) was conducted in *tert*-butyl methyl ether (TBME) at 30 °C during 18 h (Figure 1), (Table 1, entry 1). Eight commercially available lipases and two domestically prepared biocatalysts, pig liver acetone powder and wheat germ lipase, were verified as catalysts; the results are summarized in Table 1.

As shown in Table 1, immobilized lipase B from *Candida antarctica* (Novozym 435) was revealed as the best catalyst among the tested enzymes for this condensation reaction (Table 1, entry 3). The coumarin α-amino phosphonate **1** was obtained in good yield (51%) after 18 h in TBME at 30 °C. Without the enzyme, the product **1** formation was not observed (Table 1, entry 1). To confirm the promiscuous activity of CaLB in the studied reaction, non-catalytic bovine serum albumin (BSA) [45] and thermally deactivated CaLB were also tested. The obtained results indicated that only the application of BSA resulted in the formation of the target product **1** with 8% yield (Table 1, entries 20 and 21). These results clearly show that the peculiar active site of CaLB is responsible for the promotion of the studied multicomponent reaction. It is also worth noting that an attractive alternative to commercially used enzymes may be domestically made biocatalysts from readily available and cheap animal or plant tissues. The use of these biocatalysts in the model reaction led to the formation of the desired products with yields of 51% and 59%, respectively (Table 1, entries 5 and 11). The type of solvent used has a great impact on enzyme stability and activity [46]. Taking into account factors related to environmental protection, it was decided to investigate selected media commonly recognized as green and sustainable solvents in chemical processes (Table 1, entries 13–15) [46,47,48]. Among tested solvents, product **1** provided the highest yield: 81% in tetrahydro-2-methylfuran (2-MeTHF) (Table 1, entry 13). To our delight, the target coumarin α-aminophosphonate **1** was obtained in 83% yield under solvent-free conditions. The increase of the reaction time to 24 h did not substantially affected the yield (85%) (Table 1, entry 12). Therefore, in subsequent optimization studies, the reactions were conducted under solvent-free conditions. Further, we have studied the impact of temperature on the reaction course. The elevation of the temperature to 40 °C and 50 °C resulted in a decrease in the yield of product **1**, which may be explained by the changes in the quaternary structure of the used enzyme (Table 1, entries 17 and 18) [49]. Then, we also investigated the amount of catalyst. The change of the amount of enzyme from 50 mg to 80 mg led to a slight increase in the yield (Table 1, entry 19). Thus, 80 mg of Novozym 435 was found to be optimal. Analogous to other addition reactions catalyzed by lipases, the obtained chiral products remain racemic [50]. 

Additionally, three various non-enzymatic catalysts reported in the literature as eco-friendly promoters of Kabachnik–Fields reaction [51,52,53], copper-based metal organic frameworks, copper(I) iodide under microwave irradiation, and Amberlyst 15, were tested for the studied transformation, resulting in the formation of the desired product **1** with up to 42% yield (Table 1, entries 22, 23 and 25). Moreover, recently reported boron trifluoride diethyl etherate [29] was also engaged as a catalyst in studied reaction; however, product **1** was obtained with a moderate yield 31% (Table 1, entry 24). 

The reusability of the enzyme is an important economic aspect resulting in a significant reduction in the cost of the synthesis. In this work, Novozym 435 was recycled up to five times with a gradual decrease in the yield of target product 1 to 67% after the fifth cycle (Figure 2).

Finally, we applied the developed enzymatic protocol for various aldehydes, obtaining coumarin α-amino phosphonate derivatives **2**–**13** with good to excellent yields (Figure 3). The elaborated procedure provides straightforward and direct access to α-amino phosphonates, which are structural analogs of compounds with well recognized antimicrobial activity [29]. The enzymatic Kabachnik–Fields reaction with aromatic aldehydes possessing electron-donating groups as well as hetero-aromatic aldehydes provided products **2**, **3**, **10**, and **11** with the highest yield ranges, from 81% to 92% (Figure 3). A decrease in the reaction yield was observed for sterically bulky aldehydes with an electron-withdrawing nitro group located at the *meta*-position of the phenyl ring and naphthyl group, which resulted in the formation of products **8** and **12** with 51% and 33% yields, respectively (Figure 3). It is also worth noting that replacing the CF_3_- group with a methyl group in the coumarin scaffold significantly reduced the yield of the resulting product **13** to 64%. This may be explained by the substrate selectivity of the enzyme. In order to ensure the effective contact of the solid reactants, 2-MeTHF was used in several cases, resulting in the formation of products **5**–**9** and **12** with good reaction yields, up to 69% (Figure 3). The analytical data of all newly obtained compounds (**1**–**13**) are presented in the experimental part (Appendix A).

Further experiments were conducted to get insights on the reaction pathway. Under the elaborated conditions, 7-(benzylideneamino)-4-trifluoromethyl-2*H*-chromen-2-one (15) [54] was used together with dimethyl *H*-phosphite in the presence of Novozym 435 as a catalyst, which resulted in the formation of the target coumarin α-amino phosphonate **1** in a high yield (73%). The obtained results may suggest that the tested reaction proceeds through the formation of the appropriate imine followed by the addition of the phosphite form activated in the oxy-anion hole located in the active center of the enzyme (Figure 2). 

### 2.2. Cytotoxic Studies of the Library of Coumarin α-Aminophoshonates ***1***–***13*** and Coumarin ***14***

The noxious activity of all 14 examined agents on bacterial cells was measured after the analysis of the MIC and MBC tests (Figure 3), for which the MIC values were recorded in the range of 0.2–1.4 µg/mL, and MBC values −2–82 µg/mL for the investigated model strains K12, R2, R3 and R4) (Figure 4 and Figure 5).

Model strains of *E.coli* were plotted in all 48-well plates observed. The K12 and R2–R4 strains were treated with the investigated coumarin α-aminophoshonates. From analysis of the MIC and MBC assays, for all tested compounds, color changes were observed with different levels depending on dilutions. Bacterial strains R3 and R4 were the most susceptible to a modification with these compounds due to the increasing length of their LPS (visible dilutions of 10^−3^ corresponding to a concentration of 0.25 µM) versus strains K12 and R2 (visible dilutions of 10^−6^ corresponding to a concentration of 0.06 µM). The analyzed R4 strain was the most sensitive of all strains, probably due to the longest lipopolysaccharide chain in the bacterial membrane. In all cases, the MBC values were approximately 222 times higher than the MIC values (Figure 6), since the K12 *E. coli* strain has native LPS while the Fpg overproducer *E. coli* strain without native LPS has an inducible level of Fpg protein. This strain was used as an additional control to determine whether the process of oxidative stress occurs in bacterial cells under the modification with the analyzed compounds.

The strain with Fpg protein was used only as a control for other *E.coli* model strains to show oxidative stress, which was induced in bacterial cells.

### 2.3. The Analysis of Bacterial DNA Isolated from E. coli R2–R4 Strains Modified with Tested Coumarin α-Aminophoshonates

The results of our current studies (MIC values) and those previously obtained [55,56,57,58] revealed that coumarin α-aminophoshonates show a strong toxic effect on the analyzed *E. coli* model strains. The compounds numbered **1**–**4**, **6**, **8**, **9**, and **12**–**14** showed the highest activity, similar to the biological activity of commonly used antibiotics. (see Appendix A)

Modified bacterial DNA was digested with Fpg, as described earlier [59,60,61,62,63]. All selected coumarin α-aminophoshonates (Figure 7) can strongly change the topology of bacterial DNA. After digestion with Fpg, mainly in enzymes of the base excision repair system (BER), an oxidative damage of approximately 3.5% was identified, which very strongly indicates oxidative damage in bacterial DNA [64,65], particularly in the R4 strain, as evidenced by the obtained MIC and MBC values. The obtained results for individual compounds were statistically significant at the level of *p* < 0.05. (Table 2).

The results of our studies show that newly synthesized compounds can potentially be engaged as “replacements” for the currently used antibiotics in hospital and clinical infections. 

The development and synthesis of new coumarin derivatives may constitute a new alternative to the commonly used antimicrobial agents in clinical infections (Figure 8 and Figure 9). Thus, studies of the antibacterial activity of newly synthesized compounds are of great importance in nosocomial or clinical infections [66,67]. The toxic effect of the newly obtained compounds was tested on *Escherichia coli* cells of the K12 strain (having native LPS) and R2–R4 strains (with different LPS lengths). The results of the MIC and MBC tests indicate that the studied coumarin derivatives significantly influenced the defragmentation of the membrane and the structure of the bacteria cell wall containing LPS of various lengths, causing oxidative stress. This was further verified through the digestion by the specific enzyme Fpg of modified bacterial DNA (Labjot, New England Biolabs, UK), recognizing oxidized guanine and adenine. The oxidative damage observed after digestion with the Fpg protein in the bacterial DNA was compared with that obtained after the treatment with antibiotics such as ciprofloxacin, bleomycin, and cloxacillin [56,57,58]. The results of our research show that coumarin α-aminophoshonates can be used in the future as typical “substitutes” for new antimicrobial drugs. 

To study oxidative stress, the modified bacterial DNA was digested with a marker of oxidative stress: Fpg protein from the group of repair glycosylases [59,60]. We investigated whether the resulting modifications in bacterial DNA would introduce oxidative damage to the DNA chain by changing the topological three forms of bacterial DNA: ccc, oc, and linear forms. Such results were in accordance with our previous studies [56,57,58,59]. The modification of bacterial DNA with the studied compounds (Appendix A with the action of Fpg) led to the observation that all analyzed coumarin α-aminophoshonates can strongly affect the topology of bacterial DNA, even after digestion with Fpg protein based on agarose gels. The sensitivity of *E. coli* strains to the cytotoxic effect of the compounds used, and after the digestion of the Fpg protein was as follows: R4 > R2 > R3 > K12, which remains in agreement with our previous studies [55,56,57,58,59]. The obtained results show that the coumarin derivatives are highly cytotoxic towards bacterial DNA, probably by the modification of the components of the bacterial membrane and the LPS located in it, which may induce specific enzymes from topoisomerase and helicase groups, leading to the destabilization of the exposed DNA base structure. Such destabilization is crucial for the bacterial cell’s survival and may play an important role in changing its electrokinetic potential, expressing the reversal of burdens. Blocking these enzymes stops DNA replication, causing the bacterial cell’s apoptosis. In the future, cytotoxicity studies will be also carried out using various cell lines and cultures to assess the biocompatibility of test compounds with active coumarin α-aminophoshonates. The dysfunction of bacterial membranes containing different lengths of LPS in model bacterial strains is an ideal model to assess the effectiveness of these compounds in relation to the antibiotics used [60,61,62].

## 3. Materials and Methods

### 3.1. Microorganisms and Media

*E. coli* K-12 and R1–R4 strains were received from Prof. Jolanta Łukasiewicz at the Ludwik Hirszfeld Institute of Immunology and Experimental Therapy (Polish Academy of Sciences, Warsaw, Poland). The reference bacterial strains of *E.coli* (K12 ATCC 25404, R2 ATCC 39544, R3 ATCC 11775, and R4 ATCC 39543) were provided by LGC Standards U.K., and *E.coli* Fpg Y9070L was used according to the recommendation of ISO 11133: 2014. These strains were used to test the antibacterial activity of the synthesized agents. Bacteria were cultivated in a tryptic soy broth (TSB; Sigma-Aldrich, Saint Louis, MI, USA) liquid medium and on agar plates containing TSB medium at 25 °C. Alternatively, TSB agar plates were used. The specific growth rate (μ) according to first-order kinetics was measured using a microplate reader (Thermo, Multiskan FC, Vantaa, Finland) at 605 nm in TSB medium. Lanes 1kb-ladder and Quick Extend DNA ladder (New England Biolabs, Ipswich, MA, USA) were used for the MIC and MBC tests, as described in detail in the previous work [55,56,57,58], and analyzed by the Tukey test, indicated by (*p* < 0.05): * *p* < 0.05, ** *p* < 0.1, *** *p* < 0.01.

### 3.2. Minimum Inhibitory Concentration (MIC) and Minimum Bactericidal Concentration (MBC)

The MIC was estimated by a microtiter plate method using sterile 48- or 96-well plates [56]. First, precursor and TIL solutions were prepared in DMSO at 20 mg mL^−1^. Fifty microliters of the solutions were placed in the first row of the plate. Next, 25 μL of sterile TSB medium was added to the other wells, and serial dilutions were performed. Then, 200 μL of inoculated TSB medium containing resazurin (0.02 mg/mL) as an indicator was added to all the wells. TSB medium was inoculated with 10(6) colony-forming units (CFU) mL^−1^ (approximately 0.5 McFarland units) of the bacterial strains. The plates were incubated at 30 °C for 24 h. Colour changes from blue to pink or yellowish with turbidity were taken as positive, and the lowest concentration at which there was no visible color change was the MIC. The MBC was estimated based on the measurement of the dehydrogenases activity in the cultures after 24 h incubation without the ILs. Four milliliters of a dense culture (approximately 109 CFU mL^−1^) that was incubated for 24 h in TSB medium at 25 °C was added to identical test tubes. Next, the tested compounds were added to the test tubes until the mixture reached final concentrations of 10–250 mg mL^−1^. Then, the cultures containing the TILs were incubated for 1 h at 30 °C. Next, 0.1 g of CaCO_3_ and 0.1 mL of a 3% triphenyltetrazolium chloride (TTC) solution were added to the test tubes. Then, the test tubes were sealed with parafilm and incubated for 1 h at 30 °C in darkness. The lowest concentration at which there was no visible red color (formazan) was taken as the MBC.

### 3.3. Chemicals

All reagents and the solvents were purchased from Sigma-Aldrich. All solvents were of analytical grade and were used without prior distillation. All specific strains, such as *Pseudomonas fluorescens* (PfL) (catalogue number 534730, Lot. number MKBH1198V), *Candida rugosa* (CrL) (catalogue number 90860, Lot. number BCBH7102V), *Candida cylindracea* (CcL) (catalogue number 62316, Lot. number 1336707), *Burkholderia cepacian* (*Pseudomonas cepacia* lipase PcL) (catalogue number 534641, Lot. Number MKBV0029V), and *Rhizomucor miehei* (RmL) (catalogue number 80484), and bovine serum albumin were purchased from Sigma-Aldrich. Immobilized lipase from *Candida antarctica* B (Novozym 435) (catalogue number LC200223) was purchased from Novo Nordisk. Lipase from porcine pancreas, Type II (PpL) (catalogue number L-3126, Lot. number 108H1379) was purchased from Sigma-Aldrich. Pig liver was converted to the acetone powder (PLAP) by the method of Connors et al. [68]. Homemade lipase from wheat germ (WgL) was prepared according to the literature protocol [69]. Merck silica gel plates 60 F254 were used for TLC (Thin Layer Chromatography) analysis. Crude reaction mixtures were purified using column chromatography on Merck silica gel 60/230–400 mesh, with an appropriate mixture of hexane and ethyl acetate as an eluent. Nuclear magnetic resonance spectra (NMR) were performed on the Varian apparatus (Varian, Saint Louis, MI, USA) (400 MHz); the mass spectrometer was from Waters Company, Milford, CT, USA. Chemical shifts are expressed in ppm and coupling constant (*J*) in Hz, using TMS as an internal standard. High-resolution mass spectra were acquired on a Maldi SYNAPT G2-S HDMS (Waters) apparatus with a QqTOF analyser. Enzymatic reactions were performed in a vortex (Heidolph Promax 1020) equipped with incubator (Heidolph Inkubator 1000). To prove the ability of the established protocol, each reaction was repeated at least three times.

### 3.4. General Procedure for the Synthesis of Coumarin α-Aminophosphonates ***1***–***13***

The general procedure for the enzyme-catalyzed Kabachnik–Fields reaction follows: a mixture of the corresponding 7-amino-4-trifluoromethyl coumarin or 7-amino-4-methyl coumarin (1 mmol), the corresponding aldehyde (1 mmol), dimethyl phosphite (1 mmol) and Novozym 435 (80 mg) (2-Me-THF (2 mL) if noted; see Figure 2) was shaken at 200 rpm at 30 °C for 18 h. After the completion of reaction, the catalyst was separated on a glass frit funnel. The residue was washed with ethyl acetate. The combined organic phase was concentrated under a vacuum. The resulting residue was purified by column chromatography (silica gel, eluent: ethyl acetate/hexanes, 6:4) to afford the target coumarin α-aminophosphonates 1–13. The yields of the derivatives are shown in Figure 2. The structures of the products were identified by their ^1^H, ^31^P and ^13^C NMR in electronic support information (ESI) and for known compounds compared with literature data. 

**Dimethyl ((phenyl)((2-oxo-4-(trifluoromethyl)-2H-chromen-7-yl)amino)methyl) phosphonate (1).** Compound 1 was obtained according to the General method with 86% (367 mg, 0.86 mmol) as a yellow solid; m.p.: 198–199 °C, ^1^H NMR (400 MHz, CDCl_3_) *δ* 7.45 (ddd, *J* = 8.9, 6.0, 2.1 Hz, 3H), 7.40–7.29 (m, 3H), 6.67 (dd, *J* = 8.9, 2.4 Hz, 1H), 6.44 (dd, *J* = 10.2, 1.6 Hz, 2H), 5.85 (t, *J* = 8.8 Hz, 1H), 4.80 (dd, *J* = 24.2, 7.7 Hz, 1H), 3.80 (d, *J* = 10.9 Hz, 3H), 3.44 (d, *J* = 10.6 Hz, 3H); ^13^Ca NMR (100 MHz, CDCl_3_) *δ* 158.9, 156.6, 150.4, 143.2, 141.6, 134.1, 129.1, 129.0, 127.6, 124.3, 112.3, 99.7, 55.9, 54.3, 53.7, 53.7; ^31^P NMR (162 MHz, CDCl_3_) *δ* 23.6. HRMS (ESI) calcd. for C_19_H_17_F_3_NO_5_PNa [M + Na]^+^, 450.0694, found 450.0689.**Dimethyl ((4-methoxyphenyl)((2-oxo-4-(trifluoromethyl)-2H-chromen-7-yl)amino) methyl)phosphonate (2).** Compound 2 was obtained according to the General method with 92% (421 mg, 0.92 mmol) as a yellow solid; m.p.: 207–209 °C; ^1^H NMR (400 MHz, CHCl_3_) *δ* 7.44–7.40 (m, 1H), 7.37 (dd, *J* = 8.8, 2.3 Hz, 2H), 6.89 (d, *J* = 8.4 Hz, 2H), 6.64 (dd, *J* = 8.9, 2.4 Hz, 1H), 6.43 (d, *J* = 8.4 Hz, 2H), 5.76 (t, *J* = 8.6 Hz, 1H), 4.75 (dd, *J* = 23.7, 7.6 Hz, 1H), 3.79 (s, 3H), 3.77 (d, *J* = 10.6 Hz, 3H), 3.46 (d, *J* = 10.6 Hz, 3H); ^13^C NMR (100 MHz, CDCl_3_) *δ* 159.9, 156.6, 150.6, 150.5, 128.9, 128.8, 126.1, 125.8, 125.8, 114.6, 114.5, 112.3, 104.8, 99.7, 55.3, 54.2, 53.7, 53.6, 52.0; ^31^P NMR (162 MHz, CDCl_3_) *δ* 23.7. HRMS (ESI) calcd. for C_20_H_19_F_3_NO_6_PNa [M + Na]^+^, 480.0800, found 480.0794.**Dimethyl ((4-methylphenyl)((2-oxo-4-(trifluoromethyl)-2H-chromen-7-yl)amino) methyl)phosphonate (3).** Compound 3 was obtained according to the General method with 84% (371 mg, 0.84 mmol) as a yellow solid; m.p.: 194–196 °C; ^1^H NMR (400 MHz, CDCl_3_) *δ* 7.43 (dd, *J* = 8.9, 2.0 Hz, 1H), 7.33 (dd, *J* = 8.2, 2.3 Hz, 2H), 7.17 (d, *J* = 8.0 Hz, 2H), 6.64 (dd, *J* = 8.9, 2.4 Hz, 1H), 6.43 (d, *J* = 8.0 Hz, 2H), 5.66 (t, *J* = 8.8 Hz, 1H), 4.76 (dd, *J* = 23.9, 7.6 Hz, 1H), 3.78 (d, *J* = 10.8 Hz, 3H), 3.46 (d, *J* = 10.6 Hz, 3H), 2.32 (s, 3H); ^13^C NMR (100 MHz, CDCl_3_) *δ* 159.9, 156.6, 150.8, 150.7, 138.5, 131.1, 131.0, 129.8, 129.7, 127.7, 127.6, 126.0, 112.3, 109.6, 109.5, 104.7, 99.7, 55.6, 54.3, 54.2, 53.7, 52.0, 21.1; ^31^P NMR (162 MHz, CDCl_3_) *δ* 23.7. HRMS (ESI) calcd. for C_20_H_19_F_3_NO_5_PNa [M + Na]^+^, 464.0851, found 464.0847.**Dimethyl ((4-fluorophenyl)((2-oxo-4-(trifluoromethyl)-2H-chromen-7-yl)amino) methyl)phosphonate (4).** Compound 4 was obtained according to the General method with 78% (347 mg, 0.78 mmol) as a yellow solid; m.p.: 182–184 °C; ^1^H NMR (400 MHz, CDCl_3_) *δ* 7.44 (ddt, *J* = 8.8, 5.5, 2.1 Hz, 3H), 7.12–6.99 (m, 2H), 6.65 (dd, *J* = 8.9, 2.4 Hz, 1H), 6.48–6.37 (m, 2H), 5.94 (t, *J* = 8.8 Hz, 1H), 4.80 (dd, *J* = 24.1, 7.7 Hz, 1H), 3.79 (d, *J* = 10.8 Hz, 3H), 3.51 (d, *J* = 10.6 Hz, 3H); ^13^C NMR (100 MHz, CDCl_3_) *δ* 159.8, 156.6, 150.5, 150.3, 130.0, 129.5,129.4, 129.3, 126.2, 116.3, 116.2, 116.1, 116.0, 112.1, 110.0, 104.9, 99.7, 60.3, 54.2, 54.1, 53.8, 53.7; ^31^P NMR (162 MHz, CDCl_3_) *δ* 23.3. HRMS (ESI) calcd. for C_19_H_16_F_4_NO_5_PNa [M + Na]^+^, 468.0600, found 468.0596.**Dimethyl ((4-bromophenyl)((2-oxo-4-(trifluoromethyl)-2H-chromen-7-yl)amino) methyl)phosphonate (5).** Compound 5 was obtained according to the General method with 69% (349 mg, 0.69 mmol) as a yellow solid; m.p.: 188–190 °C; ^1^H NMR (400 MHz, CDCl_3_) δ 7.49 (d, *J* = 8.4 Hz, 2H), 7.43 (dd, *J* = 8.9, 1.9 Hz, 1H), 7.37–7.29 (m, 2H), 6.63 (dd, *J* = 8.9, 2.4 Hz, 1H), 6.43 (d, *J* = 8.4 Hz, 2H), 5.84 (t, *J* = 8.8 Hz, 1H), 4.77 (dd, *J* = 24.4, 7.5 Hz, 1H), 3.80 (d, *J* = 10.9 Hz, 3H), 3.53 (d, *J* = 10.7 Hz, 3H); ^13^C NMR (100 MHz, CDCl_3_) *δ* 159.7, 156.5, 150.2, 141.4, 133.4, 132.3, 132.2, 129.3, 126.2, 122.7, 112.1, 105.0, 99.8, 55.4, 54.3, 54.2, 53.9, 53.8; ^31^P NMR (162 MHz, CDCl_3_) *δ* 22.7. HRMS (ESI) calcd. for C_19_H_16_BrF_3_NO_5_PNa [M + Na]+, 527.9799, found 527.9794.**Dimethyl ((4-iodophenyl)((2-oxo-4-(trifluoromethyl)-2H-chromen-7-yl)amino) methyl)phosphonate (6).** Compound 6 was obtained according to the General method with 58% (320 mg, 0.58 mmol) as a yellow solid; m.p.: 209–211 °C; ^1^H NMR (400 MHz, CDCl_3_) *δ* 7.69 (d, *J* = 7.8 Hz, 2H), 7.42 (dd, *J* = 8.9, 1.9 Hz, 1H), 7.21 (dd, *J* = 8.5, 2.3 Hz, 2H), 6.63 (dd, *J* = 8.9, 2.4 Hz, 1H), 6.42 (d, *J* = 7.8 Hz, 2H), 5.94 (dd, *J* = 9.7, 7.7 Hz, 1H), 4.76 (dd, *J* = 24.4, 7.6 Hz, 1H), 3.80 (d, *J* = 10.8 Hz, 3H), 3.54 (d, *J* = 10.7 Hz, 3H); ^13^C NMR (100 MHz, CDCl_3_) *δ* 159.7, 156.5, 150.4, 150.2, 138.1, 134.1, 129.5, 126.2, 123.0, 112.1, 110.0, 105.0, 99.8, 94.3, 55.5, 54.3, 54.0, 53.9, 53.8; ^31^P NMR (162 MHz, CDCl_3_) *δ* 22.7. HRMS (ESI) calcd. for C_19_H_16_F_3_INO_5_PNa [M + Na]^+^, 575.9661, found 575.9655.**Dimethyl ((4-chlorophenyl)((2-oxo-4-(trifluoromethyl)-2H-chromen-7-yl)amino) methyl)phosphonate (7).** Compound 7 was obtained according to the General method with 62% (286 mg, 0.62 mmol) as a yellow solid; m.p.: 213–214 °C; ^1^H NMR (400 MHz, CDCl_3_) δ 7.44–7.39 (m, 2H), 7.34 (d, *J* = 8.7 Hz, 2H), 6.72–6.60 (m, 1H), 6.43 (d, *J* = 8.7 Hz, 2H), 5.80 (t, *J* = 8.6 Hz, 1H), 4.79 (dd, *J* = 24.3, 7.5 Hz, 1H), 3.80 (d, *J* = 10.8 Hz, 3H), 3.53 (d, *J* = 10.8 Hz, 3H); ^13^C NMR (100 MHz, CDCl_3_) *δ* 159.7, 156.5, 150.2, 134.6, 132.8, 129.3, 129.2, 129.0, 128.9, 126.2, 112.1, 110.0, 105.0, 99.8, 55.3, 54.3, 53.9, 53.8; ^31^P NMR (162 MHz, CDCl_3_) *δ* 22.9. HRMS (ESI) calcd. for C_19_H_16_F_3_ClNO_5_PNa [M + Na]^+^, 484.0304, found 484.0299.**Dimethyl ((3-nitrophenyl)((2-oxo-4-(trifluoromethyl)-2H-chromen-7-yl)amino) methyl)phosphonate (8).** Compound 8 was obtained according to the General method with 51% (241 mg, 0.51 mmol) as a yellow solid; m.p.: 195–197 °C; ^1^H NMR (400 MHz, CDCl_3_) δ 8.36 (d, *J* = 2.2 Hz, 1H), 8.16 (dd, *J* = 8.2, 1.0 Hz, 1H), 7.82 (d, *J* = 7.6 Hz, 1H), 7.56 (t, *J* = 8.0 Hz, 1H), 7.43 (dd, *J* = 8.9, 1.9 Hz, 1H), 6.67 (dd, *J* = 8.9, 2.4 Hz, 1H), 6.44 (s, 2H), 6.28 (t, *J* = 8.7 Hz, 1H), 4.98 (dd, *J* = 24.9, 7.7 Hz, 1H), 3.86 (d, *J* = 10.9 Hz, 3H), 3.65 (d, *J* = 10.8 Hz, 3H); ^13^C NMR (100 MHz, CDCl_3_) *δ* 156.5, 150.2, 150.0, 148.7, 137.2, 137.1, 133.8, 133.7, 129.9, 129.9, 126.4, 123.5, 123.5, 122.4, 122.3, 111.9, 105.2, 99.8, 55.3, 54.3, 54.1, 53.8; ^31^P NMR (162 MHz, CDCl_3_) *δ* 21.9. HRMS (ESI) calcd. for C_19_H_16_F_3_N2O_7_PNa [M + Na]^+^, 495.0545, found 495.0541.**Dimethyl ((4-nitrophenyl)((2-oxo-4-(trifluoromethyl)-2H-chromen-7-yl)amino) methyl)phosphonate (9).** Compound 9 was obtained according to the General method with 65% (307 mg, 0.65 mmol) as a yellow solid; m.p.: 199–201 °C; ^1^H NMR (400 MHz, CDCl_3_) *δ* 8.21 (d, *J* = 8.8 Hz, 2H), 7.66 (d, *J* = 11.0 Hz, 2H), 7.47 – 7.39 (m, 1H), 6.63 (dd, *J* = 8.9, 2.4 Hz, 1H), 6.43 (d, *J* = 8.8 Hz, 2H), 6.01 (dd, *J* = 10.0, 7.4 Hz, 1H), 4.96 (dd, *J* = 25.2, 7.5 Hz, 1H), 3.83 (d, *J* = 10.8 Hz, 3H), 3.62 (d, *J* = 10.8 Hz, 3H); ^13^C NMR (100 MHz, CDCl_3_) *δ* 159.5, 156.5, 150.0, 149.9, 148.0, 142.1, 142.0, 128.6, 128.5, 124.1, 111.9, 105.3, 99.9, 55.6, 54.3, 54.2, 54.1, 54.0. ^31^P NMR (162 MHz, CDCl_3_) *δ* 21.8. HRMS (ESI) calcd. for C_19_H_16_F_3_N2O_7_PNa [M + Na]^+^, 495.0545, found 495.0538.**Dimethyl ((furan-2-yl)((2-oxo-4-(trifluoromethyl)-2H-chromen-7-yl)amino) methyl)phosphonate (10).** Compound 10 was obtained according to the General method with 76% (317 mg, 0.76 mmol) as a yellow solid; m.p.: 193–194 °C; ^1^H NMR (400 MHz, CDCl_3_) δ 7.50–7.42 (m, 1H), 7.41 (q, *J* = 1.6 Hz, 1H), 6.67 (dd, *J* = 8.9, 2.4 Hz, 1H), 6.57 (d, *J* = 2.4 Hz, 1H), 6.45 (s, 2H), 6.36 (dd, *J* = 3.4, 1.9 Hz, 1H), 5.52 (t, *J* = 8.0 Hz, 1H), 4.95 (dd, *J* = 23.5, 8.7 Hz, 1H), 3.81 (d, *J* = 10.8 Hz, 3H), 3.63 (d, *J* = 10.7 Hz, 3H); ^13^C NMR (100 MHz, CDCl_3_) *δ* 159.8, 156.6, 150.3, 150.2, 147.5, 143.1, 126.2, 112.1, 111.0, 110.9, 109.5, 105.1, 99.6, 54.1, 53.9, 53.8, 48.2; ^31^P NMR (162 MHz, CDCl_3_) *δ* 21.1. HRMS (ESI) calcd. for C_17_H_15_F_3_NO_6_PNa [M + Na]^+^, 440.0487, found 440.0482.**Dimethyl ((tiophen-2-yl)((2-oxo-4-(trifluoromethyl)-2H-chromen-7-yl)amino) methyl)phosphonate (11).** Compound 11 was obtained according to the General method with 81% (351 mg, 0.81 mmol) as a yellow solid; m.p.: 205–207 °C; ^1^H NMR (400 MHz, CDCl_3_) *δ* 7.46 (dd, *J* = 9.0, 1.9 Hz, 1H), 7.30–7.25 (m, 1H), 7.20 (td, *J* = 2.9, 2.4, 1.7 Hz, 1H), 7.00 (ddd, *J* = 5.1, 3.6, 0.7 Hz, 1H), 6.70 (dd, *J* = 8.9, 2.4 Hz, 1H), 6.58 (s, 1H), 6.45 (s, 1H), 5.79 (t, *J* = 8.2 Hz, 1H), 5.11 (dd, *J* = 23.7, 8.0 Hz, 1H), 3.81 (d, *J* = 10.8 Hz, 3H), 3.61 (d, *J* = 10.6 Hz, 3H); ^13^C NMR (100 MHz, CDCl_3_) *δ* 159.8, 156.6, 150.4, 150.3, 137.6, 137.5, 127.4, 127.0, 126.9, 126.2, 126.1, 112.1, 110.1, 110.0, 105.1, 99.8, 54.4, 54.3, 53.9, 51.7, 50.1; ^31^P NMR (162 MHz, CDCl_3_) *δ* 21.7. HRMS (ESI) calcd. for C_17_H_15_F_3_NO_5_PSNa [M + Na]^+^, 456.0258, found 456.0253.**Dimethyl ((naphthalen-1-yl)((2-oxo-4-(trifluoromethyl)-2H-chromen-7-yl)amino) methyl)phosphonate (12).** Compound 11 was obtained according to the General method with 33% (158 mg, 0.33 mmol) as a yellow solid; m.p.: 218–220 °C; ^1^H NMR (400 MHz, CDCl_3_) *δ* 7.97 (s, 1H), 7.86–7.69 (m, 3H), 7.62 (d, *J* = 8.6 Hz, 1H), 7.49–7.39 (m, 2H), 7.35 (dd, *J* = 8.9, 2.0 Hz, 1H), 6.76–6.66 (m, 1H), 6.61 (t, *J* = 8.6 Hz, 1H), 6.55 (s, 1H), 6.37 (s, 1H), 5.05 (dd, *J* = 24.3, 7.9 Hz, 1H), 3.85 (d, *J* = 10.9 Hz, 3H), 3.48 (d, *J* = 10.6 Hz, 3H); ^13^C NMR (100 MHz, CDCl_3_) *δ* 159.9, 156.5, 151.0, 150.9, 141.7, 141.4, 133.2, 131.8, 128.9, 128.9, 127.9, 127.7, 127.1, 127.0, 126.5, 126.0, 125.2, 125.1, 123.1, 120.3, 112.1, 109.5, 109.5, 104.6, 99.8, 56.0, 54.5, 54.3, 54.2, 53.8; ^31^P NMR (162 MHz, CDCl_3_) *δ* 23.4. HRMS (ESI) calcd. for C_23_H_19_F_3_NO_5_PNa [M + Na]^+^, 500.0851, found 500.0845.**Dimethyl ((phenyl)((2-oxo-4-(trifluoromethyl)-2H-chromen-7-yl)amino)methyl) phosphonate (13).** Compound 13 was obtained according to the General method with 64% (239 mg, 0.64 mmol) as a yellow solid; m.p.: 205–207 °C; ^1^H NMR (400 MHz, CDCl_3_) *δ* 7.49–7.40 (m, 2H), 7.39–7.27 (m, 4H), 6.58 (dd, *J* = 8.7, 2.4 Hz, 1H), 6.40 (s, 1H), 5.95 (s, 1H), 5.49 (dd, *J* = 9.9, 7.7 Hz, 1H), 4.80 (dd, *J* = 24.2, 7.7 Hz, 1H); ^13^C NMR (100 MHz, CDCl_3_) *δ* 161.5, 155.5, 152.6, 149.6, 149.5, 134.5, 128.9, 128.5, 128.4, 127.7, 125.4, 111.6, 111.3, 110.2, 99.7, 56.0, 54.5, 53.7, 53.6, 18.4; ^31^P NMR (162 MHz, CDCl_3_) *δ* 23.9. HRMS (ESI) calcd. for C_19_H_20_NO_5_PNa [M + Na]^+^, 396.0977, found 396.0965.

## 4. Conclusions

Here we present a novel approach to coumarin α-aminophosphonates via an enzyme-catalyzed Kabachnik–Fields reaction. This protocol is characterized by mild, metal-free reaction conditions leading to the formation of the desired products with high yields. The crucial role of solvent type on the reaction efficiency was revealed. It is also worth mentioning that the biocatalyst can be reused several times, which significantly reduces the overall cost of the synthesis. The developed protocol is solvent-free, which makes this procedure environmentally benign. The obtained coumarin derivatives were also tested as potential antimicrobial agents against selected *E. coli* strains. The antimicrobial activity was enhanced by the synergetic effect of two pharmacophores, coumarin and α-amino phosphonate. The impact of the coumarin α-aminophosphonates structure on the antimicrobial activity against model strains of *Escherichia coli* K12 and R2–R4 was investigated. The newly synthesized compounds can modify all tested *E. coli* strains (R2–R4) and their bacterial DNA, changing the spatial structure of the LPS in their cell membranes. Among the tested coumarin α-aminophosphonates, compounds **4**–**8** and **10**–**12** showed super-selectivity and exhibited the highest cytotoxic activity, comparable to or better than the antibiotics ciprofloxacin, bleomycin, and cloxacillin. 

## Data Availability

On request of those interested.

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
