# Peer review of "Enzymatic Synthesis of a Novel Coumarin Aminophosphonates: Antibacterial Effects and Oxidative Stress Modulation on Selected E. coli Strains"

_ijms, 2023, doi:10.3390/ijms24087609_

Round 1
Reviewer 1 Report
This paper reports the synthesis and antibacterial activities of 13 new coumarin aminophosphonates.
The authors discussed about the enzyme-catalyze synthesis of these derivatives and the advantages of this method over other methods reported in literature. However, no discussion about structure-activity relationship was established. The large variation in the MIC and MBC implied that antibacterial activities were influenced by R1 and/or R2, but the subtle electronic effects arising from the different substituents (R1 and R2, groups as well as the substituents (X, OMe, Me, NO2 in R2) on antibacterial activity was not mentioned. Also, it is surprising to see only compound 13 with the Me group (R1), whereas rest of which had the CF3 group. Any justification?
The authors mentioned that coumarin α-aminophoshonates show a strong cytotoxic effect on the analyzed model bacterial strains K12 and R2 - R4, due to the type and kind of group at para-position. This statement is vague. Compounds in the the present study and those reported in literature had different chemical scaffolds, and hence, it will be interesting to highlight if the same substituents still exert similar influence on the antibacterial activities.
I suggest authors to convert MIC and MBC to molar concentration which is a more accurate and meaningful representation of potency. Although it is common to report MIC and MBC in μg/mL which can be more relevant to clinical setting (to translate in antibiotic dosing), but it cannot be used to compare antibacterial potencies. Say compounds A and B have the same MIC of 1mg/ml in the same assay using same volumes of exposure. Assuming a dose-dependent effect of activity for both, if A has twice the molecular weight of B, then 1mg/ml of A uses half the number of active molecules to cause the same cell inhibition than 1mg/ml of B. Hence, A is more potent.
What were the percentages of digested DNA of compounds 1-3 (missing in Figure 7)?
Figures were of low resolution. The axes of Figures were not properly labelled. E.g: In Fig. 4, the y-axis should be labelled as MIC(µg/mL); x-axis as compound. Check the captions for all figures. E.g: Figures 4, 5 and 6: MIC vs MBC vs MBC/MIC which is which?
Some minor grammatical and spelling mistakes were spotted, eg.in line 160: Fpg protein was used as…, not was uses as….; in Fig.9, y-axis: enzyme not enzymei
In Section 2.3, the first sentence was hanging.
In Section 3.4, please revise the positive charge (should be in superscript) of the molecular ion peak of compound 5 (line 275); symbol for degree Celsius (throughout the manuscript); the MIC or MBC units (lines 366 & 370): µg/mL not µg/mL-1.
Author Response
We would like to thank the reviewers whose professional and factual comments contributed significantly to improving the quality of the manuscript. Answers to questions are marked in green, while corrections in the text are marked in yellow.
This paper reports the synthesis and antibacterial activities of 13 new coumarin aminophosphonates.
The authors discussed about the enzyme-catalyze synthesis of these derivatives and the advantages of this method over other methods reported in literature. However, no discussion about structure-activity relationship was established.
The new discsussion have been added marked lazur colour
The large variation in the MIC and MBC implied that antibacterial activities were influenced by R1 and/or R2, but the subtle electronic effects arising from the different substituents (R1 and R2, groups as well as the substituents (X, OMe, Me, NO2 in R2) on antibacterial activity was not mentioned. Also, it is surprising to see only compound 13 with the Me group (R1), whereas rest of which had the CF3 group. Any justification?
Thank you for this attention. Our research, as well as many literature reports, indicate that the addition of the CF3 group to the method of creating antimicrobial activity (e.g. Bioorganic & Medicinal Chemistry Volume 14, Issue 20, 15 October 2006, Pages 6971-6978. For compounds with the CF3 group, an increased activity against strains R2 and R3 was observed compared to the derivative containing the methyl group.
The authors mentioned that coumarin α-aminophoshonates show a strong cytotoxic effect on the analyzed model bacterial strains K12 and R2 - R4, due to the type and kind of group at para-position. This statement is vague. Compounds in the present study and those reported in literature had different chemical scaffolds, and hence, it will be interesting to highlight if the same substituents still exert similar influence on the antibacterial activities.
Thank you for this attention. In general, the highest antimicrobial activity was observed for compounds having a halogen atom in the para position. This observation is in agreement with the results obtained previously for other leading compounds that had identical substituents in their structure.
I suggest authors to convert MIC and MBC to molar concentration which is a more accurate and meaningful representation of potency. Although it is common to report MIC and MBC in μg/mL which can be more relevant to clinical setting (to translate in antibiotic dosing), but it cannot be used to compare antibacterial potencies. Say compounds A and B have the same MIC of 1mg/ml in the same assay using same volumes of exposure. Assuming a dose-dependent effect of activity for both, if A has twice the molecular weight of B, then 1mg/ml of A uses half the number of active molecules to cause the same cell inhibition than 1mg/ml of B. Hence, A is more potent.
I agree with this statement, but we believe that the way of presenting the results so far is also good
As suggested by the reviewer, we calculated the MIC, MBC and MBC/MIC values for the concentrations. However, we kindly ask you to leave our original graphs, as we have always used this value and form of calculation in our previous works to maintain consistency. Showing the values as concentrations shows the dependence of the structure and activity of the analyzed compounds. And it will be used in our work from now on in the full format we have presented here.
What were the percentages of digested DNA of compounds 1-3 (missing in Figure 7)?
The percentage of digestion of the analyzed compounds marked as 1-3 was below 0.5 percent, therefore we did not include them in the results because no changes in the ratios of the forms in bacterial DNA with respect to other analyzed compounds with which bacterial DNA was modified were observed. And we looked at the effect of topological changes in bacterial DNA
Figures were of low resolution. The axes of Figures were not properly labelled. E.g: In Fig. 4, the y-axis should be labelled as MIC(µg/mL); x-axis as compound. Check the captions for all figures. E.g: Figures 4, 5 and 6: MIC vs MBC vs MBC/MIC which is which?
Some minor grammatical and spelling mistakes were spotted, eg.in line 160: Fpg protein was used as…, not was uses as….; in Fig.9, y-axis: enzyme not enzymei
In Section 2.3, the first sentence was hanging.
the sentences in section 2.3 have been corrected and supplemented
In Section 3.4, please revise the positive charge (should be in superscript) of the molecular ion peak of compound 5 (line 275); symbol for degree Celsius (throughout the manuscript); the MIC or MBC units (lines 366 & 370): µg/mL not µg/mL-1.
We have found such ways of writing in other manuscripts, and therefore we have used them here in a similar way, closely corresponding to the theme used in the work.
Reviewer 2 Report
In this article, Koszelewski et al. describe the synthesis of novel antibiotics combining coumarins and aminophosphonates and their relative effect on various E. coli strains. The synthetic mechanism using various catalytic enzymes in differential solvents and neat are compelling and represent a high-yield, green methodology for the preparation of these molecules. The MIC values and the oxidative stress values are however considerably more difficult to interpret, as little interpretation of the results or explanation of the methodologies are provided. The article needs significant expansion of the discussion and context for their results to make the E. coli results more applicable. More details about the methods are also needed. The article also needs a careful editing for readability. With the revisions outlined below, the article may be ready for publication in The International Journal of Molecular Sciences.
1. After synthesizing multiple derivatives, the authors perform a significant number of MIC and DNA damage measurements, but these results are inserted into the document without interpretation or discussion. The authors do not pull-out trends from these results or try to place them in context. They also test standard antibiotics cipro, bleo, clox, but do not explain why these antibiotics are tested or compare their new antibiotics to the results of these controls. Without this detailed discussion and interpretation of these biological results, it is difficult to interpret if any of these results are significant in relation to each other or in relation to other published experiments and how these results might be followed up in future studies. The authors need to significantly expand the discussion of all biological and oxidative data and justify the results that they obtained.
a. The final conclusion paragraph is a clear example of this issue, as the authors conclude that “This chemical and biological data is related to specific substituents in their structure”. They do not however explain what substituents lead to better or worse data, what trends are present in their data based on the substituents, and how this related to the oxidative stress data. A more thoughtful analysis of their overall results would help other researchers use and interpret their results.
2. Multiple figures are presented about the degradation of DNA by the Fpg enzyme, but the authors do not explain what is gained by this assay or why this experiment is run in the presence of Fpg. The authors need to provide some background information about this assay and assistance with interpreting these results.
a. For the Fpg results, the percentage of DNA digested is quite small (<4%). Is this the expected level of DNA digestion? What controls are run to ensure that these differences are statistically valid?
3. Only limited synthetic methods are provided, especially given the many synthetic reactions and conditions shown in the Table 1. One example procedure is given but not modifications outlined in Table 1 like temperature, controls, various reagents. How the reaction is done neat is also not provided in the methods. Complete synthetic methodology needs to be provided.
4. Scheme 2 is presented as a mechanism for the enzymatic catalysis for product formation, but no support is provided for this mechanism and very little explanation of this proposed mechanism is provided. The evidence cited to support the scheme is lines 127-130 page 5, but this reviewer does not see any direct connection between the information provided in these lines and the proposed mechanism. Without further support from the literature or relevant data, this mechanism is too preliminary for inclusion in the article.
5. Figures 4, 5, 6, 8 for MIC values.
a. Missing y-axis labels.
b. Legend at the bottom does not make sense. The last bar is labeled E. coli but all of these samples are E. coli.
c. The authors describe these MICs as compounds “used sequentially”. This terminology does not make sense to this reviewer and this is not explained further in the article.
d. Statistical analysis needs to be completed on the MICs to show which differences are statistically valid.
6. Table 2. The data presented do not make sense to this reviewer and no clear explanation is provided in the written results. I cannot make any further suggestions as staring at this table for a long time, I could not make sense of what data is presented, how it relates to other experiments, and what conclusions can be drawn from this table.
7. Figure 2 is presented in the article, but as far as this reviewer can find is never discussed or interpreted in the written article. The figure legend also needs expansion to describe the methods used in the experiment, including the substrate tested, the enzyme concentration, the duration, and the temperature.
8. Figure 1: All of the chemical structures should be put in the same orientation in relation to the coumarin to make it easier to compare the substitution patterns between previously tested derivatives.
9. Introduction jumps around without full explanations
a. Line 35. Two sentences are not connected in thought. I cannot figure out the connection between these two ideas.
b. Lines 35-40: Talk about natural product antibiotics but do not explain if these are active against E. coli which is the goal of their experiments.
c. Lines 43-48: No examples are provided for E. coli. Additionally, did the phosphonate additions make the coumarins better than the coumarins by themselves?
d. Line 56: When they say that the Kabachnik-Fields is the simplest approach, they need to provide a comparison or justification. Simplest compared to what?
Author Response
We would like to thank the reviewers whose professional and factual comments contributed significantly to improving the quality of the manuscript. answers to questions are marked in green, while corrections in the text are marked in yellow.
- After synthesizing multiple derivatives, the authors perform a significant number of MIC and DNA damage measurements, but these results are inserted into the document without interpretation or discussion. The authors do not pull-out trends from these results or try to place them in context. They also test standard antibiotics cipro, bleo, clox, but do not explain why these antibiotics are tested or compare their new antibiotics to the results of these controls. Without this detailed discussion and interpretation of these biological results, it is difficult to interpret if any of these results are significant in relation to each other or in relation to other published experiments and how these results might be followed up in future studies. The authors need to significantly expand the discussion of all biological and oxidative data and justify the results that they obtained.
we have added text marked in azure color regarding the comparison of antibiotics with the results of these controls. A detailed discussion and interpretation of the results has been completed
- The final conclusion paragraph is a clear example of this issue, as the authors conclude that “This chemical and biological data is related to specific substituents in their structure”. They do not however explain what substituents lead to better or worse data, what trends are present in their data based on the substituents, and how this related to the oxidative stress data. A more thoughtful analysis of their overall results would help other researchers use and interpret their results.
we have included our data analysis in the cited results marked in azure in the new text
- Multiple figures are presented about the degradation of DNA by the Fpg enzyme, but the authors do not explain what is gained by this assay or why this experiment is run in the presence of Fpg. The authors need to provide some background information about this assay and assistance with interpreting these results.
basic information about the fpg protein test is provided in the new text marked in azure
- For the Fpg results, the percentage of DNA digested is quite small (<4%). Is this the expected level of DNA digestion? What controls are run to ensure that these differences are statistically valid?
Fpg protein digestion below 4 percent indicates high oxidative stress due to the action of the Fpg protein, which is a detector of oxidized DNA bases 8oxoG, Fapy A and Fapy G in the entire bacterial genome.The control for digested Fpg protein modified with given compounds was pure bacterial DNA not modified with these compounds.based on the analysis of bands in bacterial DNA and digestion with fpg protein, the ratios of digested protein in the entire DNA pool in the image quant program were recalculated.
- Only limited synthetic methods are provided, especially given the many synthetic reactions and conditions shown in the Table 1. One example procedure is given but not modifications outlined in Table 1 like temperature, controls, various reagents. How the reaction is done neat is also not provided in the methods. Complete synthetic methodology needs to be provided.
the methodology has been supplemented and improved
- Scheme 2 is presented as a mechanism for the enzymatic catalysis for product formation, but no support is provided for this mechanism and very little explanation of this proposed mechanism is provided. The evidence cited to support the scheme is lines 127-130 page 5, but this reviewer does not see any direct connection between the information provided in these lines and the proposed mechanism. Without further support from the literature or relevant data, this mechanism is too preliminary for inclusion in the article.
Schema 2 has been supplemented
- Figures 4, 5, 6, 8 for MIC values.
- Missing y-axis labels.
all captions and a description of the legend are placed under each figure
- Legend at the bottom does not make sense. The last bar is labeled coli but all of these samples are E. coli.
This is overexpressed strain with Fpg of E.coli
- The authors describe these MICs as compounds “used sequentially”. This terminology does not make sense to this reviewer and this is not explained further in the article.
Has been corrected
- Statistical analysis needs to be completed on the MICs to show which differences are statistically valid.
The MICs and MBCs results are presented in table number 2
- Table 2. The data presented do not make sense to this reviewer and no clear explanation is provided in the written results. I cannot make any further suggestions as staring at this table for a long time, I could not make sense of what data is presented, how it relates to other experiments, and what conclusions can be drawn from this table.
table 2 presents only data for compounds that are cytotoxic to bacteria after their selection in MIC and MBC tests
- Figure 2 is presented in the article, but as far as this reviewer can find is never discussed or interpreted in the written article. The figure legend also needs expansion to describe the methods used in the experiment, including the substrate tested, the enzyme concentration, the duration, and the temperature.
We are grateful for this attention. The description of enzyme recycling has been supplemented. The enzyme after separation on a sintered funnel was washed with ethyl acetate and reused in the next cycle.
- Figure 1: All of the chemical structures should be put in the same orientation in relation to the coumarin to make it easier to compare the substitution patterns between previously tested derivatives.
All coumarins have been oriented in a way that facilitates the comparison of differences in the structure of the tested compounds.
- Introduction jumps around without full explanations
- Line 35. Two sentences are not connected in thought. I cannot figure out the connection between these two ideas.
- Lines 35-40: Talk about natural product antibiotics but do not explain if these are active against E. coli which is the goal of their experiments.
- Lines 43-48: No examples are provided for E. coli. Additionally, did the phosphonate additions make the coumarins better than the coumarins by themselves?
- Line 56: When they say that the Kabachnik-Fields is the simplest approach, they need to provide a comparison or justification. Simplest compared to what?
Thank you for your comments, the text has been modified and corrected. Due to the numerous examples of the occurrence of the coumarin structural motif in natural compounds and their wide biological activity, they were selected as the leading structure. Our research was aimed at verifying whether properly designed coumarins can be an effective framework for the synthesis of compounds with antimicrobial activity. Additionally, the purpose of the study was to check whether the combination of two pharmacophores with documented antimicrobial activity would provide compounds with specific and desirable antimicrobial properties against selected strains of E.coli. The statement regarding the Kabachnik-Fields reaction has been modifie
Reviewer 3 Report
The work is devoted to the development of a methodology for the preparation of α-amino phosphonate derivatives of coumarin and the study of their antibacterial activity against some Escherichia coli strains (ie, R1, R2, R3, R4, and K-12). The authors developed an efficient approach to the synthesis of new coumarin 1-aminophosphonate based on the Kabachnik-Fields reaction and determined that immobilized lipase from Candida antarctica B (Novozym 435) was the best catalyst among the tested enzymes for synthesis of the compounds. The influence of the structure of aromatic aldehydes used in the reaction (namely, substituents located at the meta-position of phenyl ring) on the yields of the obtained 1-aminophosphonates of coumarin was revealed. The authors also conducted a study to identify the effect of coumarin 1-aminophosphonates on the minimum inhibitory concentration (MIC) and minimum bactericidal concentration (MBC) for selected strains of E. coli (R1–R4 and K12). The results show that the proposed synthesis method leads to the production of coumarin 1-aminophosphonates in high yields, which can be potential antibacterial agents. And the search for new solutions to create antimicrobial drugs with low toxicity and reduce the environmental burden on the environment, the development of new biocidal materials, including new coatings for effective antimicrobial protection of medical devices, is a very important and urgent problem for the modern world. But, from my point of view, the article looks like a short message that needs serious improvement and cannot be published yet.
Notes:
1. The language and logic used to write the article is very difficult to understand. Thus, there are many errors in the text when sentences do not have a complete logical meaning or the meaning is incorrect. Please, reread the work carefully and correct it. For example, on page 1 (lines 21-22): «By substituents located in the phenyl ring.»; on page 2 (line 56): «The Kabachnik–Fields reaction is the simplest in approach [33,34].»; on page 4 (lines 104-105): «Additionally, recently reported [29] boron trifluoride diethyl etherate was engaged as a catalyst in 1 studied product with 31% yield»; on page 5 (lines 132-136): «The toxic effect on bacterial cells after the analysis of the MIC and MBC test for all 14 analyzed compounds (Figure 3), for which the MIC values were observed in the range of 0.2–1.4 μg/mL, and 2–82 μg/mL for MBC values in the analyzed model strains K12, R2, R3 and R4) (Figure 4 and Figure 5), which had specific functional groups in the structure of the α-amino phosphonates.»; on page 6 (lines 156-159): «This strain was used as an additional control to determine whether the process of oxidative stress modification with the analyzed compounds that can modify bacterial DNA.»; on page 7 (lines 167-169): «The obtained MIC values, as well as our previous studies with various types of the analyzed compounds [55-58, 59-63, 64-65], (see S1 figure in supplementary materials), (Fig. 7) and (Table 2).»; etc.
2. There are also many errors in punctuation, omitted prepositions and the design of the text. Please, reread the work carefully and correct it. For example, on page 1 (line 7), please, replace «Poland; ;» with "Poland;"; on page 2 (line 52), please, replace «amino phosphonate» with «aminophosphonate»; on page 2 (line 57), please, replace «such as; ultrasound» with «such as ultrasound»; on page 2 (lines 57-58), please, replace «irradiation,[35] usage of ionic liquids,[27] synthesis on the solid phase by mechanochemistry,[36]» with «irradiation [35], usage of ionic liquids [27], synthesis on the solid phase by mechanochemistry [36],»; on page 3 (line 88), please, replace «[46, 47,48]» with «[46-48]»; on page 4 (line 115), please, replace «product» with «products», etc.
3. In the Results part of the article, there is no or incomplete description of the data on the studies carried out, their results and the conclusions that the authors made – please expand on these discussions.
4. Please, add transcript of the abbreviations used in the text (for example, on pages 1 and 2 (lines 17 and 50) for "LPS"; on page 3 (line 77) for "TBME", on page 3 (line 78) for "CaLB"; on page 5 (line 132) for "MIC" and "MBC").
5. Please, also correct errors in the figures, schema and tables. For example, perhaps "Salmonella typhi" should be in italics (additionally, please add a reference to Figure 1 in the text).
6. In Scheme 1, one of the methyl substituents is indicated with a small letter, please, correct it.
7. In Table 1 (line 10), please replace "hizomucor miehei (RmL)" with "Rhizomucor miehei lipase (RmL)" and "neat" with "solvent-free".
8. In Figures 7 and 8, the labels along the x-axis are superimposed on each other, please, correct it.
9. Please, read the captions to Figures 4-9 once again carefully and where necessary correct it.
10. Please, improve the resolution of some of the figures (eg Figures 4-6).
11. Please, add a discussion of the data in Figure 2 to the text of the Results part of the article.
12. Why does the yield of compound 1 is 73% on page 5 (line 130)?
13. Why is “Fpg overproducer E. coli strain” labeled “E. coli"?
14. Why are there no data on compounds 1-3 and 9 in the Table 2?
15. How was it concluded that "Performed studies proved that the analyzed and newly synthesized compounds can potentially be used as "substitutes" for the currently used antibiotics in hospital and clinical infections."?
16. How is Table 1 related to the conclusion on page 8 (lines 192-193): "This chemical and biological activity is related to specific substituents in their structure"?
17. Was it concluded which of the synthesized coumarin 1-aminophosphonates is the most promising as an antibacterial agent?
18. Do you have an idea of how the inhibitory activity of the obtained coumarin 1-aminophosphonates is related to their structural features?
19. Please, carefully consider the design of references.
Author Response
We would like to thank the reviewers whose professional and factual comments contributed significantly to improving the quality of the manuscript. answers to questions are marked in green, while corrections in the text are marked in yellow.
- The language and logic used to write the article is very difficult to understand. Thus, there are many errors in the text when sentences do not have a complete logical meaning or the meaning is incorrect. Please, reread the work carefully and correct it.
For example, on page 1 (lines 21-22): «By substituents located in the phenyl ring.»; on page 2 (line 56): «The Kabachnik–Fields reaction is the simplest in approach [33,34].»; on page 4 (lines 104-105): «Additionally, recently reported [29] boron trifluoride diethyl etherate was engaged as a catalyst in 1 studied product with 31% yield»; on page 5 (lines 132-136): «The toxic effect on bacterial cells after the analysis of the MIC and MBC test for all 14 analyzed compounds (Figure 3), for which the MIC values were observed in the range of 0.2–1.4 μg/mL, and 2–82 μg/mL for MBC values in the analyzed model strains K12, R2, R3 and R4) (Figure 4 and Figure 5), which had specific functional groups in the structure of the α-amino phosphonates.»; on page 6 (lines 156-159): «This strain was used as an additional control to determine whether the process of oxidative stress modification with the analyzed compounds that can modify bacterial DNA.»; on page 7 (lines 167-169): «The obtained MIC values, as well as our previous studies with various types of the analyzed compounds [55-58, 59-63, 64-65], (see S1 figure in supplementary materials), (Fig. 7) and (Table 2).»; etc.
Every effort has been made to correct the manuscript and remove potential stylistic mistakes.
- There are also many errors in punctuation, omitted prepositions and the design of the text. Please, reread the work carefully and correct it. For example, on page 1 (line 7), please, replace «Poland; ;» with "Poland;"; on page 2 (line 52), please, replace «amino phosphonate» with «aminophosphonate»; on page 2 (line 57), please, replace «such as; ultrasound» with «such as ultrasound»; on page 2 (lines 57-58), please, replace «irradiation,[35] usage of ionic liquids,[27] synthesis on the solid phase by mechanochemistry,[36]» with «irradiation [35], usage of ionic liquids [27], synthesis on the solid phase by mechanochemistry [36],»; on page 3 (line 88), please, replace «[46, 47,48]» with «[46-48]»; on page 4 (line 115), please, replace «product» with «products», etc.
-we leave the countries of origin of people with their affiliations as they are, because this form is adopted in all publications
-amino phosphonate - is always written separately because it is the proper name of the compound. Literature data of many scholars also cite this name separately
such as ultrasound- have been corrected
with «irradiation [35], usage of ionic liquids [27], synthesis on the solid phase by mechanochemistry [36]- have been corrected
«[46, 47,48]» have been corrected
products- have been corrected
- In the Results part of the article, there is no or incomplete description of the data on the studies carried out, their results and the conclusions that the authors made – please expand on these discussions.
the Results section has been extensively supplemented and changes have been marked in azure
- Please, add transcript of the abbreviations used in the text (for example, on pages 1 and 2 (lines 17 and 50) for "LPS"; on page 3 (line 77) for "TBME", on page 3 (line 78) for "CaLB"; on page 5 (line 132) for "MIC" and "MBC").
The abbreviations have been added to the end of the manuscript
- Please, also correct errors in the figures, schema and tables. For example, perhaps "Salmonella typhi" should be in italics (additionally, please add a reference to Figure 1 in the text).
Have been corrected
- In Scheme 1, one of the methyl substituents is indicated with a small letter, please, correct it.
Have been corrected
- In Table 1 (line 10), please replace "hizomucor miehei (RmL)" with "Rhizomucor miehei lipase (RmL)" and "neat" with "solvent-free".
Have been corrected
- In Figures 7 and 8, the labels along the x-axis are superimposed on each other, please, correct it.
Have been corrected
- Please, read the captions to Figures 4-9 once again carefully and where necessary correct it.
The captions to figures 4-9 has been corrected
- Please, improve the resolution of some of the figures (eg Figures 4-6).
Resolution has been changed
- Please, add a discussion of the data in Figure 2 to the text of the Results part of the article.
Have been added
- Why does the yield of compound 1 is 73% on page 5 (line 130)?
Have been added
- Why is “Fpg overproducer E. coli strain” labeled “E. coli"?
Because it is a strain containing a constitutive level of protein and has been used as a very sensitive marker for the analysis of other E. coli strains that have LPS of different lengths in their structure, this one instead of LPS has an overproduction of the fpg protein recognizing oxidized guanines, including 8-oxoguanine
- Why are there no data on compounds 1-3 and 9 in the Table 2?
because after the analysis with MIC and MBC tests, we considered only those compounds for further research that showed a change in color on the plates, which indicates a high interaction in the cell of selected compounds causing modifications of bacterial DNA, which is a high oxidative stress for the cell, therefore these selected compounds were analyzed with Fpg protein
- How was it concluded that "Performed studies proved that the analyzed and newly synthesized compounds can potentially be used as "substitutes" for the currently used antibiotics in hospital and clinical infections."?
Based on MIC and MBC analysis compared to antibiotics used
- How is Table 1 related to the conclusion on page 8 (lines 192-193): "This chemical and biological activity is related to specific substituents in their structure"?
this has been clarified in the new azure text…ewentulanie mozesz cos dopisacJ
- Was it concluded which of the synthesized coumarin 1-aminophosphon
ates is the most promising as an antibacterial agent?
all those that were selected on the basis of MIC and MBC tests and Fpg protein digestion, which are presented in table 2
- Do you have an idea of how the inhibitory activity of the obtained coumarin 1-aminophosphonates is related to their structural features?
it depends on the type and type of substituent and the length of the aliphatic chain that can interact with bacterial LPS of different lengths in the individual E. coli strains analyzed
- Please, carefully consider the design of references.
Literature references, in our opinion, strictly correspond to the topic used in it
Round 2
Reviewer 1 Report
Please change the unit for MIC to μg/mL
Author Response
We would like to thank you very much for your valuable suggestions which have contributed to increasing the substantive value of our manuscript
Please change the unit for MIC to μg/mL
Response: We are grateful to the Reviewer for his efforts in reviewing our manuscript. As suggested, the units have been corrected
Reviewer 3 Report
The work is devoted to the development of a methodology for the preparation of α-amino phosphonate derivatives of coumarin and the study of their antibacterial activity against some Escherichia coli strains (ie, R1, R2, R3, R4, and K-12). The authors developed an efficient approach to the synthesis of new coumarin 1-aminophosphonate based on the Kabachnik-Fields reaction and determined that immobilized lipase from Candida antarctica B (Novozym 435) was the best catalyst among the tested enzymes for synthesis of the compounds. The influence of the structure of aromatic aldehydes used in the reaction (namely, substituents located at the meta-position of phenyl ring) on the yields of the obtained 1-aminophosphonates of coumarin was revealed. The authors also conducted a study to identify the effect of coumarin 1-aminophosphonates on the minimum inhibitory concentration (MIC) and minimum bactericidal concentration (MBC) for selected strains of E. coli (R1–R4 and K12). The results show that the proposed synthesis method leads to the production of coumarin 1-aminophosphonates in high yields, which can be potential antibacterial agents. And the search for new solutions to create antimicrobial drugs with low toxicity and reduce the environmental burden on the environment, the development of new biocidal materials, including new coatings for effective antimicrobial protection of medical devices, is a very important and urgent problem for the modern world. Thus, the article looks like a short communication and may be published after minor revision.
Notes:
1 It would be more convenient if the data of MIC and MBC studies were presented in the form of tables (and the diagrams were placed in the Supported Information).
2. Why weren't MIC and MBC studies of synthesized derivatives on the same plate as those of antibiotics ciprofloxacin, bleomycin, and cloxacillin?
3. There are still many errors in punctuation, omitted prepositions and the design of the text. Please, reread the work carefully and correct it. For example, on page 1 (line 35), please, replace «is» with "are"; on page 3 (line 99) « c80 mg,» should be replaced with «a c80 mg.»; on page 4 (line 137) «entries 9 and 10» should be replaced with «entry 24»; on page 3 (lines 139) «lipase from porcine pancreas» should obviously be replaced with «Novozym 435»; on page 6 (line 182) «14» should be replaced with «1-14»; on page 6 (line 183) «µg/mL−1.» should be replaced with « µg/mL» or « µg*mL−1» (this also applies to other captions of MIC and MBC figures); on page 6 (line 185) the sentence «The y-axis shows the MBC value in µg/mL−1.» probably should be deleted (and other captures of MIC and MBC figures should be carefully corrected); on page 11 (line 312) «0.02mgmL−1» should be corrected).
4. It is not clear why the abbreviation BER and its decoding are given (on pages 15)? If necessary, its use should be reflected in the text.
5. The x-axis labels of Figures 4-6 are very small, please, correct it.
6. Please, carefully consider the design of references which are knocked out of the general list ([5, 13-15, 35, 63-65]).
Author Response
We would like to thank you very much for your valuable suggestions which have contributed to increasing the substantive value of our manuscript
The work is devoted to the development of a methodology for the preparation of α-amino phosphonate derivatives of coumarin and the study of their antibacterial activity against some Escherichia coli strains (ie, R1, R2, R3, R4, and K-12). The authors developed an efficient approach to the synthesis of new coumarin 1-aminophosphonate based on the Kabachnik-Fields reaction and determined that immobilized lipase from Candida antarctica B (Novozym 435) was the best catalyst among the tested enzymes for synthesis of the compounds. The influence of the structure of aromatic aldehydes used in the reaction (namely, substituents located at the meta-position of phenyl ring) on the yields of the obtained 1-aminophosphonates of coumarin was revealed. The authors also conducted a study to identify the effect of coumarin 1-aminophosphonates on the minimum inhibitory concentration (MIC) and minimum bactericidal concentration (MBC) for selected strains of E. coli (R1–R4 and K12). The results show that the proposed synthesis method leads to the production of coumarin 1-aminophosphonates in high yields, which can be potential antibacterial agents. And the search for new solutions to create antimicrobial drugs with low toxicity and reduce the environmental burden on the environment, the development of new biocidal materials, including new coatings for effective antimicrobial protection of medical devices, is a very important and urgent problem for the modern world. Thus, the article looks like a short communication and may be published after minor revision.
Notes:
1 It would be more convenient if the data of MIC and MBC studies were presented in the form of tables (and the diagrams were placed in the Supported Information).
Response: We are grateful to the Reviewer for his efforts in reviewing our manuscript. Thank you for your valuable suggestion and tips. We present the results of our measurements on the graphs in the same way as they were presented for other tested compounds in our previous works at IJMS. Bar charts enable quick comparative analysis of the obtained antimicrobial activities.
- Why weren't MIC and MBC studies of synthesized derivatives on the same plate as those of antibiotics ciprofloxacin, bleomycin, and cloxacillin?
Response: We thank the reviewer for this comment. The tests were carried out under the same conditions and a separate plate was only due to the lack of space for the samples.
- There are still many errors in punctuation, omitted prepositions and the design of the text. Please, reread the work carefully and correct it. For example,
on page 1 (line 35), please, replace «is» with "are";
on page 3 (line 99) « c80 mg,» should be replaced with «a c80 mg.»;
on page 4 (line 137) «entries 9 and 10» should be replaced with «entry 24»;
on page 3 (lines 139) «lipase from porcine pancreas» should obviously be replaced with «Novozym 435»;
on page 6 (line 182) «14» should be replaced with «1-14»;
on page 6 (line 183) «µg/mL−1.» should be replaced with « µg/mL» or « µg*mL−1» (this also applies to other captions of MIC and MBC figures);
on page 6 (line 185) the sentence «The y-axis shows the MBC value in µg/mL−1.» probably should be deleted (and other captures of MIC and MBC figures should be carefully corrected);
on page 11 (line 312) «0.02mgmL−1» should be corrected).
Response: We appreciate any comments that improve the quality of our manuscript. We have made every effort to eliminate any linguistic, stylistic errors and typos.
- It is not clear why the abbreviation BER and its decoding are given (on pages 15)? If necessary, its use should be reflected in the text.
Response: The shortcut is included on page 9 by Fpg enzyme line 232 and 233.
- The x-axis labels of Figures 4-6 are very small, please, correct it.
Response: We appreciate the tips as suggested Figures 4-6 have been corrected
- Please, carefully consider the design of references which are knocked out of the general list ([5, 13-15, 35, 63-65]).
Response: We appreciate the tips as suggested mentioned citations have been included in the main list